# Co-Expression of Androgen Receptor and Cathepsin D Defines a Triple-Negative Breast Cancer Subgroup with Poorer Overall Survival

**DOI:** 10.3390/cancers12051244

**Published:** 2020-05-15

**Authors:** Hanane Mansouri, Lindsay B. Alcaraz, Caroline Mollevi, Aude Mallavialle, William Jacot, Florence Boissière-Michot, Joelle Simony-Lafontaine, Valérie Laurent-Matha, Pascal Roger, Emmanuelle Liaudet-Coopman, Séverine Guiu

**Affiliations:** 1IRCM (Institut de Recherche en Cancérologie de Montpellier), INSERM (Institut National de la Santé et de la Recherche Médicale), Univ Montpellier (University of Montpellier), ICM (Institut du Cancer de Montpellier), 34298 Montpellier, France; mansouri-hanane@outlook.fr (H.M.); lindsay.alcaraz@inserm.fr (L.B.A.); aude.mallavialle@inserm.fr (A.M.); william.jacot@icm.unicancer.fr (W.J.); valerie.matha@inserm.fr (V.L.-M.); pascal.roger@chu-nimes.fr (P.R.); severine.guiu@icm.unicancer.fr (S.G.); 2Biometry Department, ICM (Institut du Cancer de Montpellier), 34298 Montpellier, France; caroline.mollevi@icm.unicancer.fr; 3Department of Medical Oncology, ICM (Institut du Cancer de Montpellier), 34298 Montpellier, France; 4Translational Research Unit, ICM (Institut du Cancer de Montpellier), 34298 Montpellier, France; Florence.Boissiere@icm.unicancer.fr (F.B.-M.); j.simony@sfr.fr (J.S.-L.); 5Department of Pathology, CHU (Centre Hospitalier Universitaire) Nîmes, 30029 Nîmes, France

**Keywords:** androgen receptor, cathepsin D, triple-negative breast cancer, prognosis

## Abstract

Background: In the triple-negative breast cancer (TNBC) group, the luminal androgen receptor subtype is characterized by expression of androgen receptor (AR) and lack of estrogen receptor and cytokeratin 5/6 expression. Cathepsin D (Cath-D) is overproduced and hypersecreted by breast cancer (BC) cells and is a poor prognostic marker. We recently showed that in TNBC, Cath-D is a potential target for antibody-based therapy. This study evaluated the frequency of AR/Cath-D co-expression and its prognostic value in a large series of patients with non-metastatic TNBC. Methods: AR and Cath-D expression was evaluated by immunohistochemistry in 147 non-metastatic TNBC. The threshold for AR positivity (AR+) was set at ≥1% of stained cells, and the threshold for Cath-D positivity (Cath-D+) was moderate/strong staining intensity. Lymphocyte density, macrophage infiltration, PD-L1 and programmed cell death (PD-1) expression were assessed. Results: Scarff-Bloom-Richardson grade 1–2 and lymph node invasion were more frequent, while macrophage infiltration was less frequent in AR+/Cath-D+ tumors (62.7%). In multivariate analyses, higher tumor size, no adjuvant chemotherapy and AR/Cath-D co-expression were independent prognostic factors of worse overall survival. Conclusions: AR/Cath-D co-expression independently predicted overall survival. Patients with TNBC in which AR and Cath-D are co-expressed could be eligible for combinatory therapy with androgen antagonists and anti-Cath-D human antibodies.

## 1. Introduction

Triple-negative breast cancers (TNBC) are defined by the lack of estrogen receptor (ER), progesterone receptor (PR) and HER2 (Human Epidermal Receptor 2) expression/amplification. They represent 15% of all breast cancers (BC). Despite surgery, adjuvant chemotherapy, and radiotherapy, prognosis in patients with TNBC is poor, mainly due to the disease heterogeneity and absence of targeted therapies. Among the seven TNBC subtypes defined on the basis of their gene expression profile [1], the luminal androgen receptor subgroup is distinct from basal-like tumors. This subtype is characterized by the lack of ER and cytokeratin 5/6 (CK5/6) expression, and the expression of genes that are usually detected in ER positive (ER+) luminal tumors, such as androgen receptor (AR) [1]. By immunohistochemistry (IHC), AR is detected in 70–90% of ER+ BC, and less frequently in TNBC with variable results among studies (8–58%), possibly due to the heterogeneity of the used assays and of the positivity cut-offs [2]. In TNBC, the prognostic value of AR expression is controversial [3,4,5,6]. In a previous study, we showed that patients with AR-positive (AR+) TNBC have poorer prognosis and higher risk of late relapse, particularly when tumors co-express AR and FOXA1 (Forkhead box A1), a transcription factor expressed by luminal tumors and acting as an AR co-activator [7]. Moreover, AR silencing in three cell lines representative of the luminal androgen receptor subtype (including MDA-MB-453) significantly reduced cell growth and colony formation, indicating that AR is partly responsible for tumor cell viability/survival [1]. Recent studies suggest that the AR antagonists bicalutamide [8], enzalutamide [9], and abiraterone acetate [10] have an antitumoral effect in AR+/ER negative (ER-) BC.

Human cathepsin D (Cath-D) is a ubiquitous, lysosomal, aspartic endoproteinase that is proteolytically active at low pH. Human Cath-D is synthesized as a 52-kDa precursor that is converted into the active 48-kDa single-chain intermediate in endosomes, and then into the fully active mature form, composed of a 34-kDa heavy chain and a 14-kDa light chain, in lysosomes. Cath-D is overproduced and aberrantly secreted by human epithelial BC cells [11]. In BC, Cath-D promotes cancer cell proliferation [11,12,13], fibroblast outgrowth [14,15], tumor angiogenesis [16,17], tumor growth, and metastasis formation [12]. Several independent clinical studies have shown that Cath-D level in the cytosol of primary BC is an independent prognostic marker correlated with the incidence of clinical metastases and shorter overall survival [18,19]. A meta-analysis of 11 studies on node-negative BC [20] and a study on 2810 BC in Rotterdam [21] confirmed that high Cath-D level is a marker of BC aggressiveness. Recently, our group showed that Cath-D is a tumor-specific extracellular marker in TNBC, and validated the feasibility and efficacy of an immunomodulatory antibody-based strategy against Cath-D to treat patients with TNBC [22]. Interestingly, the fully human anti-Cath-D F1 antibody triggers not only natural killer cell activation, but also prevents the tumor recruitment of immunosuppressive M2 tumor-associated macrophages (TAMs) and myeloid-derived suppressor cells (MDSC) in TNBC cell and patient-derived xenografts, a specific effect associated with a less immunosuppressive tumor microenvironment [22]. 

In BC, TAMs are the most abundant inflammatory cells and are typically M2-polarized with suppressive capacity [23], linked to their enzymatic activities and production of anti-inflammatory cytokines [24]. Recent evidences indicate that they support tumor progression and metastasis formation by blocking the anti-tumor immunity and by secreting factors that promote angiogenesis and epithelial-to-mesenchymal transition re-activation [23]. High TAM levels have been associated with poorer BC outcome [25]. 

Programmed cell death (PD-1) (a member of the CD-28-CTLA-4 family) is an immune check-point receptor expressed by immune cells that contributes to the immune tolerance of self-antigens by peripheral T cells. PD-L1 (one of its ligand) is expressed by immune cells, epithelial BC cells, and tumor-infiltrating lymphocytes (TILs). Activation of the PD-1-PD-L1 pathway specifically inhibits T-cell activation and is one of the mechanisms that allow cancer cells to escape the antitumor immune response [26]. TNBC are thought to be more immunogenic than other BC. Indeed, the available evidence indicates that in TNBC, PD-L1 expression is more frequent (up to 60%) than in other BC, and that PD-L1 tumor expression is positively associated with stromal TILs [27]. The prognostic value of PD-L1 expression in TNBC is still controversial, possibly due to the lack of standardized assays (different antibodies, scoring systems, positivity thresholds, and tumor microenvironment compartment included in the analysis) [27]. 

The aim of the present study was to evaluate AR and Cath-D expression profiles, the prognostic value of AR/Cath-D co-expression, and its correlation with PD-L1 expression, TIL density, and macrophage infiltration in a large series of patients with non-metastatic TNBC and long follow-up. The objective was to identify a TNBC subgroup with worse prognosis and eligible for combinatory therapy with androgen antagonists and anti-Cath-D antibodies.

## 2. Materials and Methods

### 2.1. Patients

This study included 147 patients with unifocal, unilateral, non-metastatic TNBC who underwent surgery at Montpellier Cancer Institute between 2002 and 2012. TNBC samples were provided by the Biological Resource Center (Biobank number BB-0033-00059) after approval by the Montpellier Cancer Institute Institutional Review Board, following the French Ethics and Legal dispositions for the patients’ information and consent. All patients were informed before surgery that their surgical specimens might be used for research purposes. Patients did not receive neoadjuvant chemotherapy before surgery. ER and PR negativity were defined as <10% expression by IHC, and HER2 negativity was defined as IHC 0/1+ or 2+ and negative fluorescent/chromogenic hybridization in situ. AR status was available for all patients. This study was reviewed and approved by the Montpellier Cancer Institute Institutional Review Board (ID number ICM-CORT-2016-04).

### 2.2. Construction of Tissue Microarrays

Tumor tissue blocks with enough material at gross inspection were selected from the Biological Resource Center. After hematoxylin-eosin-safranin (HES) staining, the presence of tumor tissue in sections was evaluated by a pathologist. Two representative tumor areas, to be used for the construction of the tumor microarrays (TMAs), were identified on each slide. A manual arraying instrument (Manual Tissue Arrayer 1, Beecher Instruments, Sun Prairie, WI, USA) was used to extract two malignant cores (1 mm in diameter) from the two selected areas. When possible, normal breast epithelium was also sampled as internal control. After arraying completion, 4 μm sections were cut from the TMA blocks. One section was stained with HES and the others were used for IHC.

### 2.3. Tumor Microarray (TMA) Immunohistochemistry

TMA sections were incubated with antibodies against Cath-D (mouse monoclonal, clone C-5, Santa Cruz Technology, Dallas, TX, USA) CK5/6 (mouse monoclonal, clone 6D5/16 B4, Dako Agilent, Santa Clara, CA, USA), EGFR (mouse monoclonal, clone 31G7, inVitroGen, Carlsbad, CA, USA), AR (mouse monoclonal, clone AR441, Dako), PD-1 (mouse monoclonal, clone MRQ-22, BioSB, Santa Barbara, CA, USA), PD-L1 (rabbit monoclonal, clone SP142, Roche Diagnostics, Bâle, Switzerland), and CD163 (mouse monoclonal, clone 10D6, BioSB) on a Autostainer Link48 platform (Dako) using the Flex® system for signal amplification and diaminobenzidine tetrahydrochloride as chromogen. TMA sections were analyzed independently by two trained observers both blinded to the clinicopathological characteristics and patient outcomes. In case of disagreement, sections were revised by a third observer to reach a consensus. Results from the duplicate cores, when available, were averaged. Basal-like phenotype was defined by CK5/6+ and/or EGFR+ (>10% tumor cells). AR positivity cut-off was ≥1% (nuclear staining) (Figure 1A,B). Cath-D was quantified in cancer cells according to the staining intensity (0: none, +: weak, ++: moderate and +++: strong). The scores 0 and + corresponded to Cath-D-negative (Cath-D-) tumors, and the scores ++ and +++ to Cath-D-positive (Cath-D+) tumors (Figure 1C). Examples of AR-/Cath-D- and of AR+/Cath-D+ tumors are shown in Figure 1A and Figure 1B, respectively. TIL density (peritumoral and intratumoral) was evaluated on HES-stained sections, and was scored as: 0 (no TILs), 1 (rare TILs), 2 (moderate infiltrate, fewer TILs than tumor cells), 3 (diffuse infiltrate, more TILs than tumor cells). PD-1 expression (IHC) by TILs was scored as follows: not evaluable (no TILs), 0 (no stained TIL), 1 (<10% of stained TILs), 2 (10–50% of stained TILs), and 3 (> 50% of stained TILs). PD-L1 expression by tumor cells was considered positive if detected in ≥1% of cells. TAM density was scored on CD163 stained sections and compared to TIL density: 0 (no macrophage), 1 (rare macrophages), 2 (moderate infiltrate, fewer macrophages than TILs), 3 (diffuse infiltrate, more macrophages than TILs).

### 2.4. Triple-Negative Breast Cancer (TNBC) Cell Lines, Enzalutamide Treatment, and Western Blotting

The MDA-MB-453 and MDA-MB-468 cell lines were obtained from SIRIC Montpellier Cancer. The SUM159 cell line was from Asterand Bioscience (Royston, U.K. & Detroit, MI, USA). The MDA-MB-231 cell line was previously described [12]. MDA-MB-231, MDA-MB-453 and MDA-MB-468 cells were cultured in DMEM with 10% fetal calf serum (FCS) (Eurobio scientific, Les Ulis, France). SUM159 cells were cultured in RPMI with 10% FCS. Cells at 90% confluence were incubated with DMSO (vehicle; control) or with enzalutamide (20 µM; Selleckchem, Munich, Germany) in the absence of FCS for 24 h. The corresponding conditioned media were centrifuged at 500× *g* for 5 min. Cell lysates were prepared in lysis buffer (50 mM Hepes (pH7.5), 150 mM NaCl, 10% glycerol, 1% Triton X-100, 1.5 mM MgCl_2_, 1 mM EGTA) containing cOmplete™ Protease Inhibitor Cocktail (Roche, Switzerland) and centrifuged at 13,000× *g* for 10 min. For western blotting experiments, proteins from cell lysates (30 µg) and conditioned media (40 µL) were separated on 13.5% SDS-PAGE and analyzed by immunoblotting with the mouse monoclonal anti-Cath-D (#610801; BD Transduction Laboratories^TM^, San Jose, CA, USA) (to detect cellular Cath-D), rabbit polyclonal anti-Cath-D (H-75; sc-10725; Santa Cruz Biotechnology, Dallas, TX, USA) (to detect secreted Cath-D), and rabbit polyclonal anti-β actin (#A2066, Sigma-Aldrich, Saint-Louis, MO, USA) antibodies using standard techniques.

### 2.5. Statistical Analyses

Data were described using medians and ranges for continuous variables, and frequencies and percentages for categorical variables. Comparisons were performed with the Kruskal-Wallis test (continuous variables) and the chi-square or Fisher’s exact test, if appropriate (categorical variables). All tests were two-sided, and *p*-values < 0.05 were considered as significant. The median follow-up was calculated using the reverse Kaplan-Meier method. Relapse-free survival (RFS) and overall survival (OS) were estimated using the Kaplan-Meier method and compared with the log-rank test. RFS was defined as the time between the date of the first histology and the date of the first recurrence at any site. Surviving patients without recurrence and patients lost to follow-up were censored at the time of the last follow-up or last documented visit. OS was defined as the time between the date of the first histology and the date of death from any cause. Multivariate analyses were performed using the Cox proportional hazard model. Hazard ratios (HR) were given with their 95% confidence interval (95% CI). All statistical analyses were performed with the STATA 13.0 software (StatCorp, College Station, TX, USA).

## 3. Results

### 3.1. Patient Characteristics 

This study included 147 women with TNBC specimens that were used to construct the tumor microarrays (TMAs) (Table 1). Their median age was 61.6 years (range 30.2–98.6) and 68% of them received adjuvant chemotherapy (ACT). Most tumors (53.1%) were pT2 and 61.2% pN0. Moreover, 86.2% were ductal carcinomas, 6.9% lobular carcinomas, and 6.9% other histological types; 11% of tumors were Scarff-Bloom-Richardson (SBR) grade 1–2. A basal-like phenotype was observed in 61.6% of samples; 66% of tumors expressed PD-L1.

### 3.2. Androgen Receptor (AR) Expression

AR expression was detected in 107 TNBC (72.8%). Comparison of the clinical and tumor characteristics in function of the tumor AR status showed that tumor size was smaller (*p* = 0.044), and lymph node involvement was more frequent (47.9% vs. 25%; *p* = 0.036) in AR+ (*n* = 107, 72.8%) than with AR− (*n* = 40, 27.2%) TNBCs (Table 1). Moreover, SBR grade was lower (SBR 1–2: 14.1% vs. 2.6%; *p* = 0.048) and Cath-D expression in tumor cells more frequent (87.3% vs. 72.5%; *p* = 0.035) in AR+ than AR− tumors. Similarly, macrophage infiltration was less important in AR+ tumors (*p* = 0.036). TIL density, PD-L1 expression on tumor cells and PD-1 expression on TILs were not significantly different between AR+ and AR− tumors.

### 3.3. AR and Cath-D Co-Expression

Cath-D expression was available for 142 TNBC samples (Table 1). Patients with AR+/Cath-D+ tumors (*n* = 89, 62.7%) had significantly more frequent lymph node involvement (46.1% vs. 28.3%; *p* = 0.036), and a trend to lower histological grade (SBR grades 1–2: 13.6% vs. 3.8%; *p* = 0.062) than patients with TNBC that did not co-express AR and Cath-D (Figure 1; Table 2). Moreover, macrophage infiltration was less frequent in AR+/Cath-D+ (*p* = 0.041). TIL density, PD-L1 expression on tumor cells, and PD-1 expression on TILs were not different. 

### 3.4. Survival Analyses

The median follow-up time was 5.4 years (range 0.1–14.3). Local or regional recurrence occurred in 10 (7%) patients, and metastatic recurrence (alone or with loco-regional recurrence) in 32 (22.5%) patients. There was a trend for lower recurrence-free survival (RFS) in patients with AR+/Cath-D+ tumors (*p* = 0.097): the 3-year RFS rates were 67.4% (CI 95% (54.1–77.6)) and 81.9% (CI 95% (68.0–90.1)) for AR+/Cath-D+ TNBCs and the other TNBCs, respectively (Figure 2). 

The 5-year RFS rates were 57.6% (CI 95% (43.0–69.7)) and 71.4% (CI 95% (55.4–82.5)) for AR+/Cath-D+ TNBC and the other TNBCs, respectively. In univariate analyses, tumor size, nodal status and ACT were statistically correlated with RFS (Table 3). Patients with tumors displaying higher TIL density tended to have a better RFS (*p* = 0.054). However, in multivariate analyses, only ACT was an independent prognostic factor of RFS (HR = 0.40, 95% CI (0.21–0.74), *p* = 0.004) (Table 3).

During the follow-up, 45 (31.7%) patients died among whom 11 (7.7%) were without a diagnosis of recurring BC. Overall survival (OS) tended to be lower in patients with AR+/Cath-D+ tumors (*p* = 0.086): the 3-year OS rates were 85.1% (CI 95% (75.8–91.1)) and 90.4% (CI 95% (78.5–95.9)) for AR+/Cath-D+ TNBCs and the other TNBCs, respectively (Figure 3; Appendix A).

In univariate analyses, age (*p* = 0.013), tumor size (*p* = 0.002), nodal status (*p* = 0.004), and ACT (*p* = 0.004) were significantly associated with OS (Table 4). In multivariate analyses, higher tumor size (*p* = 0.024), no ACT (*p* < 0.001), and AR/Cath-D co-expression (*p* = 0.034) were independent prognostic factors of worse OS (Table 4).

### 3.5. Effect of the AR Inhibitor Enzalutamide on Cath-D Expression and Secretion in TNBC Cells

As our multivariate analysis indicated that OS was worse in patients with AR+/Cath-D+ TNBC, combination treatment with an AR antagonist and the human anti-Cath-D F1 antibody [22] may be of interest. Therefore, due to the estrogen-like effect of AR [28], the estrogen-regulation of Cath-D [29,30] and the inhibition of Cath-D secretion by ER antagonists [31,32], we wanted first to determine whether anti-androgen treatment affects Cath-D expression or secretion in AR+/Cath-D+ TNBC cells. To this aim, we tested the AR antagonist enzalutamide in four different AR+/Cath-D+ TNBC cell lines (SUM159, MDA-MB-231, MDA-MB-453, and MDA-MB-468 cells) [33] that express and secrete Cath-D (Figure 4; Appendix A). Incubation with enzalutamide (20 µM) for 24 h did not affect Cath-D expression or secretion (Figure 4).

## 4. Discussion

In our study, with cut-offs of 1% for AR positivity, 72.8% of TNBC expressed AR. This percentage is higher than in other studies [2], except in the work by Traina et al. where nuclear AR expression was higher than 0% in nearly 80% of all evaluable samples [9]. We chose the cut-off of 1% for AR because a recent clinical trial used this threshold and demonstrated the clinical activity of an AR antagonist [9]. Moreover, recent data suggest that AR-targeted therapies may enhance chemotherapy efficacy even in TNBC with low AR expression by targeting cancer stem cell-like cells [34]. While most studies have shown that AR expression is a good prognostic factor in ER+ tumors [35,36,37], it is more controversial for ER-tumors where AR signaling could drive tumor growth [38]. Indeed, it was suggested that in TNBC, AR might use estrogen response element-like motifs to bind to DNA and induce transcription of genes that regulate cell growth in an ER-independent manner [28]. In a meta-analysis based on retrospective studies on TNBC and on population data, AR positivity was significantly associated with prolonged RFS, but had no significant impact on OS [6]. Conversely, in the Nurses’ Health Studies, AR was associated with improved BC survival in patients with ER+/HER2− tumors and with worse survival in patients with TNBC (*n* = 4147 women with BC, including 581 with TNBC) [5]. In the first 7 years post-diagnosis, AR expression was associated with a 62% increase in BC-specific mortality in patients with ER-tumors after adjustments for patient, tumor, and treatment covariates [5]. Battharai et al. evaluated AR prognostic value in TNBC from six international cohorts (*n* = 1407) and found that AR status alone was not a reliable prognostic marker [4]. In our study there was no significant association between AR expression and RFS or OS. These results underline that prospective data are needed to conclude on AR prognostic significance in TNBC and that another biomarker is required to identify a subgroup with worse prognosis in this specific population.

Cath-D, p53, and BCL-2, assessed by IHC, are prognostic indicators of BC metastatic spreading [39]. Cath-D quantified by cytosolic assay in primary BC is a well-established marker of poor prognosis independently of the ER status [20,21]. In ER+ BC cell lines, estrogen and growth factors stimulate Cath-D protein and mRNA accumulation [30,40]. The regulation of Cath-D mRNA accumulation by estrogen is mainly through transcription initiation increase [29]. Estrogen responsive elements have been detected in the proximal promoter region of the Cath-D-encoding gene *CTSD* [41]. High Cath-D level (by IHC) has been associated with poorer prognosis in patients with ER+ BC, including tumors of lower histological grade [42,43,44]. This suggests that Cath-D may identify a subgroup of more aggressive tumors. Recently, Cath-D expression was assessed in cytosols of different primary BC subtypes (ER^+^/HER2^+^, ER^−^/HER2^+^, ER^+^/HER2^−^, and ER^−^/HER2^−^) using a cytosolic assay [22]. The mean Cath-D level in the TNBC subtype was in the range of the cut-off values reported in clinical studies on all combined BC subtypes [20,21]. A recent IHC study found that Cath-D was overexpressed in 71.5% of TNBC analyzed (*n* = 504) and proposed a prognostic model for TNBC outcome based on node status, Cath-D expression, and Ki67 index [45]. In addition, high *CTSD* mRNA expression was significantly associated with shorter RFS in a cohort of 255 patients with TNBC [22], suggesting that Cath-D overexpression might be a predictive marker of poor TNBC prognosis. However, Cath-D prognostic value had not been studied in AR+ TNBC before. Taking into account all these observations including the ER-like activity of AR in TNBC and the estrogen-mediated regulation of Cath-D, here we assessed the prognostic value of AR/Cath-D co-expression in TNBC. 

In our study, Cath-D expression in tumor cells was more frequently detected in AR+ than AR-TNBC, and 62.7% of non-metastatic TNBC harbored AR/Cath-D co-expression. To our knowledge, this is the first study to investigate the profile and the prognostic value of this association in TNBC. AR+/Cath-D+ TNBC seemed to behave like luminal tumors, with a morphological profile distinct from that of other TNBC. Moreover, patients with an AR+/Cath-D+ tumors had a higher risk of relapse and a significant worse OS than patients with other TNBC types. Importantly, in our study, AR/Cath-D co-expression was an independent prognostic factor for OS, but not AR or Cath-D expression on its own, underlying the importance of their co-expression. Moreover, although lymph node involvement was more common in AR+/Cath-D+ tumors in our study, nodal status was not an independent prognostic factor. Usually, in TNBC, relapses occur in the first 3 years of follow-up. This was confirmed in our study for patients with AR- or AR+/Cath-D-tumors. On the other hand, relapses occurred even after a longer interval in patients with AR+/Cath-D+ tumors, like in patients with ER+/HER2− tumors. Prospective data in large cohorts are needed to confirm the prognostic value of AR/Cath-D co-expression in TNBC. 

For patients with AR+/Cath-D+ TNBC at risk of late relapse, an adjuvant anti-androgen therapy could be considered, like for ER+ tumors. In the metastatic setting, the clinical benefit rate of anti-androgens (delivered as monotherapy) is only about 20% [8,9,10]. Therefore, new combinations of targeted therapies are urgently needed in this TNBC subgroup. In patients with metastatic or locally advanced TNBC, the atezolizumab (anti-PD-L1 antibody) plus nab-paclitaxel combination prolonged progression-free survival in the entire population and in the PD-L1+ subgroup, in a randomized phase III study [46]. Interestingly, among patients with PD-L1+ tumors, the median OS was 25.0 months with atezolizumab and 15.5 months with chemotherapy alone [46]. A clinical trial is currently assessing a selective androgen receptor modulator and an anti-PD-1 antibody in patients with metastatic AR+ TNBC (NCT02971761). 

In agreement with the literature [27], 66.9% of TNBC expressed PD-L1. PD-L1 expression on tumor cells and PD-1 expression on TILs were not different in tumor co-expressing or not AR and Cath-D, in our population. Thus, AR and Cath-D co-expression does not allow defining a subgroup of patients who could benefit most from the combination of anti-androgens and check-point inhibitors. 

We recently showed that extracellular Cath-D could be considered as a biomarker in TNBC and a therapeutic target for the fully human anti-Cath-D F1 antibody [22]. Treatment with the F1 antibody of mice xenografted with MDA-MB-231 TNBC cells led to tumor depletion of pro-tumoral M2-polarized TAMs and of MDSCs [22]. In addition, co-culture assays showed that mesenchymal stem cell homing towards MDA-MB-231 cells depends on the chemoattractive effect of extracellular Cath-D [47]. Thus, the association of anti-androgen therapy and anti-Cath-D immunotherapy may be of interest in TNBC. Here, we observed that TAM density was lower in AR+/Cath-D+ tumors. High TAM density has been associated with poor survival rates in BC [25] and also with negative hormone receptor status and malignant phenotype [25]. Thus, the anti-Cath-D F1 antibody might reduce the level of M2-TAMs, allowing the re-activation of immune cells. Similarly, treatment of BC explants with the matrix metalloproteases inhibitor BB-94 reduced tumor growth in mice, not by directly targeting tumor cells, but by indirectly decreasing the number of recruited TAMs, possibly through inhibition of their mesenchymal migration properties [48]. As estradiol antagonists inhibit Cath-D secretion [31,32], we confirmed in vitro that treatment with an androgen antagonist does not affect Cath-D expression or secretion in AR+/Cath-D+ TNBC cells before testing the possibility of combination therapy using anti-androgens and anti-Cath-D antibodies. These data suggest the feasibility of the association of anti-androgens and the F1 antibody to treat patients with AR+/Cath-D+ tumors. This is in agreement with previous studies [49,50] suggesting that AR inhibition, especially in combination with immunotherapy, may provide a potential novel therapeutic option for selected patients with TNBC.

## 5. Conclusions

In this series, almost 63% of TNBC co-expressed AR and Cath-D and displayed distinct clinicopathological characteristics. AR/Cath-D co-expression independently predicted OS. Patients with AR+/Cath-D+ tumors tended to have higher risk of late recurrences than patients with other TNBC types. These biomarkers could be useful to identify a specific TNBC subgroup with worse prognosis. Our results could have therapeutic implications because anti-androgens are under investigation and anti-Cath-D antibodies are tested in pre-clinical studies.

## Figures and Tables

**Figure 1 cancers-12-01244-f001:**
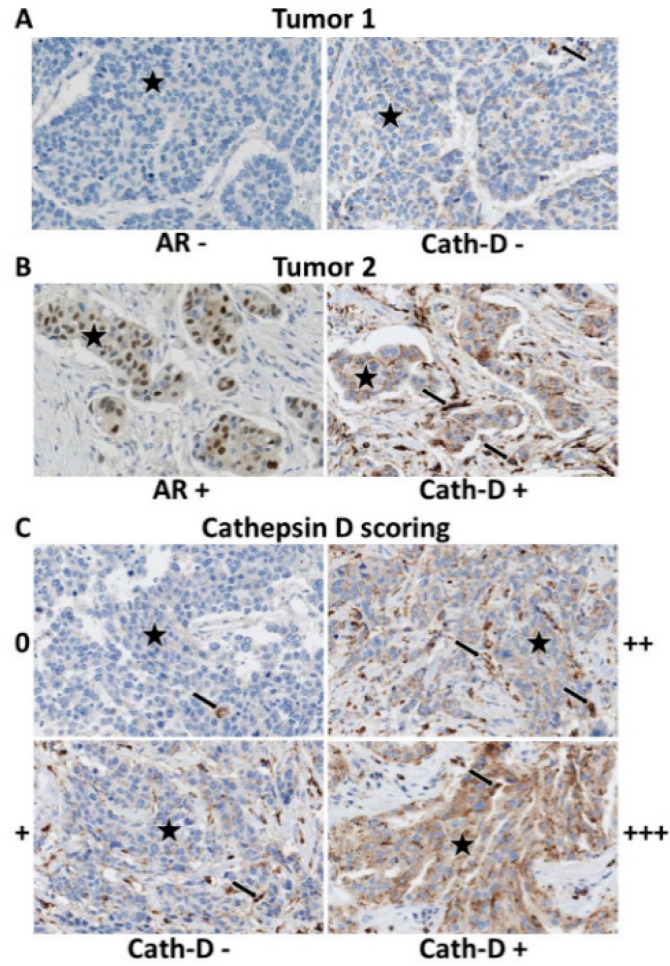
Representative images of TNBC that express or not AR and Cath-D. Immunohistochemistry analysis of TNBC tumor microarrays (TMAs). (**A**) AR-/Cath-D− tumor. (**B**) AR+/Cath-D+ tumor. (**C**) Cath-D scoring in TNBC cancer cells: 0 (none), + (weak), ++ (moderate), +++ (strong). Magnification ×200. Stars: tumor cells; arrows: macrophages.

**Figure 2 cancers-12-01244-f002:**
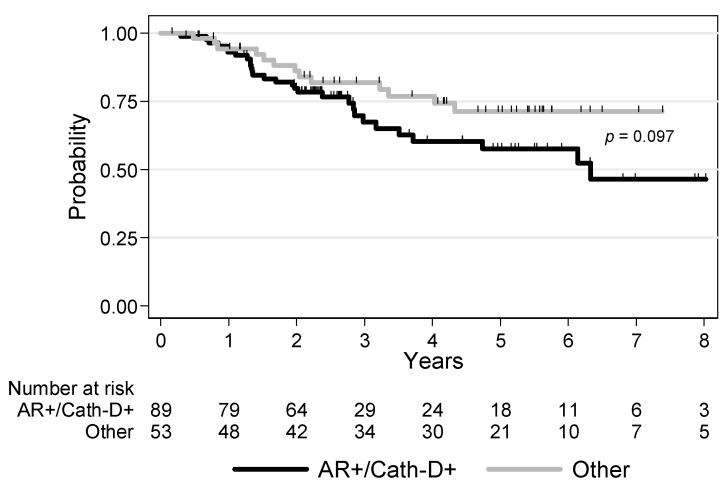
Recurrence-free survival in patients with non-metastatic TNBC (*n* = 142) in function of AR and Cath-D co-expression.

**Figure 3 cancers-12-01244-f003:**
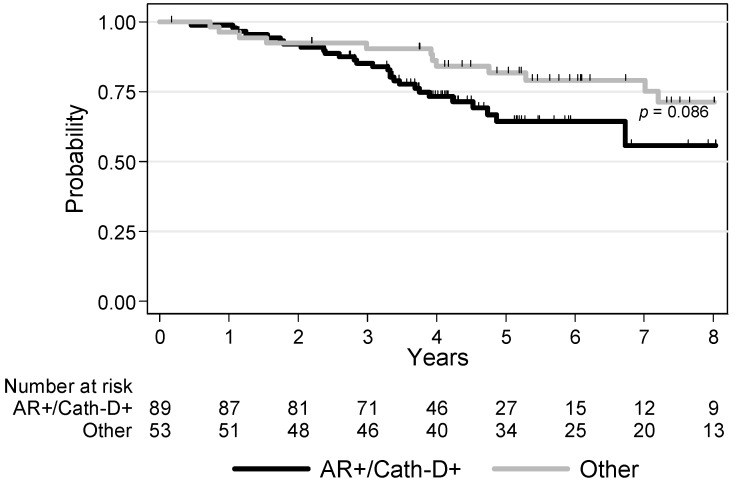
Overall survival in patients with non-metastatic TNBC (*n* = 142) in function of AR and Cath-D co-expression.

**Figure 4 cancers-12-01244-f004:**
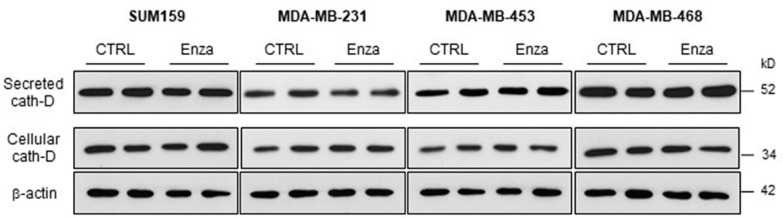
Effect of the AR inhibitor enzalutamide on Cath-D expression and secretion in TNBC cells. AR-positive TNBC cells were incubated with DMSO (control) or with enzalutamide (20 µM) in the absence of fetal calf serum (FCS) for 24 h. Whole cell extracts (30 µg proteins) and conditioned media (40 µL) were separated by 13.5% SDS-PAGE. Cellular Cath-D and secreted Cath-D were detected with the appropriate anti-Cath-D antibodies. β-actin, loading control. kD, molecular mass in kiloDaltons.

**Table 1 cancers-12-01244-t001:** Clinical and tumor characteristics of the whole population and according to the triple-negative breast cancer (TNBC) androgen receptor (AR) status.

Clinical and Tumor Characteristics	Total Population *n* = 147	AR+ TNBC *n* = 107 (72.8%)	AR− TNBC *n* = 40 (27.2%)	*p* Value
**Age (years), median [min–max]** <55 years ≥55 years	61.6 [30.2–98.6] 50 (34.0%) 97 (66.0%)	61.4 [30.2–98.6] 36 (33.6%) 71 (66.4%)	63.2 [30.8–86.3] 14 (35.0%) 26 (65.0%)	0.877
**Tumor size** T1 T2 T3/T4	52 (35.4%) 78 (53.1%) 17 (11.5%)	44 (41.1%) 53 (49.5%) 10 (9.4%)	8 (20.0%) 25 (62.5%) 7 (17.5%)	**0.044**
**Node status** N− N+	90 (61.2%) 57 (38.8%)	60 (56.1%) 47 (47.9%)	30 (75.0%) 10 (25.0%)	**0.036**
**Histological grade (SBR)** 1–2 3	16 (11.0%) 129 (89.0%)	15 (14.1%) 91 (85.9%)	1 (2.6%) 38 (97.4%)	**0.048**
**Histology** Ductal Lobular Other	124 (86.2%) 10 (6.9%) 10 (6.9%)	89 (85.6%) 9 (8.7%) 6 (5.7%)	35 (87.5%) 1 (2.5%) 4 (10.0%)	0.312
**Adjuvant chemotherapy** No Yes	47 (32.0%) 100 (68.0%)	34 (31.8%) 73 (68.2%)	13 (32.5%) 27 (67.5%)	0.933
**Basal-like phenotype** Yes No	90 (61.6%) 56 (38.4%)	64 (60.4%) 42 (39.6%)	26 (65.0%) 14 (35.0%)	0.608
**TIL density** [0–2] 3	105 (74.5%) 36 (25.5%)	73 (73.3%) 28 (27.7%)	32 (80.0%) 8 (20.0%)	0.343
**PDL-1 expression in tumor cells** <1% ≥1%	45 (33.1%) 91 (66.9%)	28 (29.2%) 68 (70.8%)	17 (42.5%) 23 (57.5%)	0.132
**PD-1 expression in TILs** 0 1 2 3	18 (13.0%) 29 (20.9%) 74 (53.1%) 18 (13.0%)	13 (13.0%) 20 (20.0%) 58 (58.0%) 9 (9.0%)	5 (12.8%) 9 (23.1%) 16 (41.0%) 9 (23.1%)	0.115
**Cath-D expression in tumor cells** No (score 0/+) Yes (score ++/+++)	24 (16.9%) 118 (83.1%)	13 (12.7%) 89 (87.3%)	11 (27.5%) 29 (72.5%)	**0.035**
**Macrophages (inflammation)** 0/+1 2 3	25 (17.6%) 44 (31.0%) 73 (51.4%)	18 (17.5%) 38 (36.9%) 47 (45.6%)	7 (18.0%) 6 (15.3%) 26 (66.7%)	**0.036**

*p* Value in bold, statistically significant.

**Table 2 cancers-12-01244-t002:** Clinical and tumor characteristics of the whole population and according to the AR/Cath-D co-expression status.

Clinical and Tumor Characteristics	Total Sample *n* = 147	AR+/Cath-D+ TNBC *n* = 89 (62.7%)	Other TNBC Types *n* = 53 (37.3%)	*p* Value
**Age (years), median [min–max]** <55 years ≥55 years	61.6 [30.2–98.6] 50 (34.0%) 97 (66.0%)	61.6 [30.2–98.6] 28 (31.5%) 61 (68.5%)	60.7 [30.8–86.3] 20 (37.7%) 33 (62.3%)	0.445
**Tumor size** T1 T2 T3/T4	52 (35.4%) 78 (53.1%) 17 (11.5%)	37 (41.6%) 44 (49.4%) 8 (9.0%)	13 (24.5%) 32 (6.4%) 8 (15.1%)	0.101
**Node status** N− N+	90 (61.2%) 57 (38.8%)	48 (53.9%) 41 (46.1%)	38 (71.7%) 15 (28.3%)	**0.036**
**Histological grade (SBR)** 1–2 3	16 (11.0%) 129 (89.0%)	13 (13.6%) 76 (86.4%)	2 (3.8%) 50 (96.2%)	0.062
**Histology** Ductal Lobular Other	124 (86.2%) 10 (6.9%) 10 (6.9%)	75 (86.2%) 6 (6.9%) 6 (6.9%)	45 (86.5%) 3 (5.8%) 4 (7.7%)	0.955
**Adjuvant chemotherapy** No Yes	47 (32.0%) 100 (68.0%)	28 (31.5%) 61 (68.5%)	17 (32.1%) 36 (67.9%)	0.939
**Basal-like phenotype** Yes No	90 (61.6%) 56 (38.4%)	53 (60.2%) 35 (39.8%)	35 (66.0%) 18 (34.0%)	0.490
**TIL density** [0–2] 3	105 (74.5%) 36 (25.5%)	65 (76.5%) 20 (23.5%)	39 (73.6%) 14 (26.4%)	0.702
**PDL-1 expression in tumor cells** <1% ≥1%	45 (33.1%) 91 (66.9%)	23 (28.1%) 59 (71.9%)	21 (39.6%) 32 (60.4%)	0.161
**PD-1 expression in TILs** 0 1 2 3	18 (13.0%) 29 (20.9%) 74 (53.1%) 18 (13.0%)	12 (14.3%) 16 (19.0%) 47 (56.0%) 9 (10.7%)	5 (9.6%) 12 (23.1%) 26 (50.0%) 9 (17.3%)	0.556
**Macrophages** 0/+1 2 3	25 (17.6%) 44 (31.0%) 73 (51.4%)	15 (17.4%) 32 (37.2%) 39 (45.4%)	9 (17.7%) 9 (17.7%) 33 (64.6%)	**0.041**

*p* Value in bold, statistically significant.

**Table 3 cancers-12-01244-t003:** Univariate and multivariate analyses for recurrence-free survival (RFS).

Clinical and Tumor Characteristics	Univariate Analysis HR 95% CI	Multivariate Analysis HR 95% CI
**Age** <55 years ≥55 years	1 1.68 [0.83–3.41] *p* = 0.138	
**Tumor size** T1 T2 T3/T4	1 1.66 [0.74–3.74] 4.69 [1.88–11.7] ***p* = 0.004**	1.47 [0.63–3.40] 3.34 [1.20–9.28] *p* = 0.056
**Node status** N− N+	1 2.67 [1.43–4.98] ***p* = 0.002**	1 1.90 [0.94–3.85] *p* = 0.074
**Histological grade (SBR)** 1–2 3	1 0.80 [0.36–1.82] *p* = 1.82	
**Histology** Ductal Lobular Other	1 1.50 [0.59–3.84] 0.39 [0.05–2.83] *p* = 0.371	
**Adjuvant chemotherapy** No Yes	1 0.41 [0.22–0.75] ***p* = 0.005**	1 0.40 [0.21–0.74] ***p* = 0.004**
**Basal-like phenotype** Yes No	1 1.63 [0.89–2.98] *p* = 0.117	
**AR** AR+ AR−	1 0.57 [0.27–1.20] *p* = 0.122	
**AR/Cath-D co-expression** AR+/Cath-D+ Other profiles	1 0.58 [0.30–1.12] ***p* = 0.097**	1 0.56 [0.28–1.12] *p* = 0.093
**TIL density** [0–2] 3	1 0.46 [0.19–1.09] *p* = 0.054	
**PDL-1 expression in tumor cells** <1% ≥1%	1 0.74 [0.39–1.40] *p* = 0.360	
**PD-1 expression in TILs** 0 1 2 3	1 1.30 [0.41–4.08] 1.23 [0.42–3.57] 0.80 [0.2–3.20] *p* = 0.821	
**Cath-D expression in tumor cells** No (score 0/+) Yes (score ++/+++)	1 1.44 [0.61–3.43] *p* = 0.386	
**Macrophages** 0/+1 2 3	1 1.98 [0.78–4.97] 1.09 [0.43–2.74] *p* = 0.148	

HR = hazard ratio; CI = confidence interval; *p* Value in bold, statistically significant.

**Table 4 cancers-12-01244-t004:** Univariate and multivariate analyses for overall survival (OS).

Clinical and Tumor Characteristics	Univariate Analysis HR 95% CI	Multivariate Analysis HR 95% CI
**Age** <55 years ≥55 years	1 2.52 [1.12–5.66] ***p* = 0.013**	
**Tumor size** T1 T2 T3/T4	1 3.02 [1.16–7.86] 6.17 [2.11–18.0] ***p* = 0.002**	1 2.31 [0,86–6.23] 4.61 [1.46–14.5] ***p* = 0.024**
**Node status** N− N+	1 2.38 [1.31–4.33] ***p* = 0.004**	1 1.82 [0.92–3.58] *p* = 0.082
**Histological grade (SBR)** 1–2 3	1 1.05 [0.44–2.88] *p* = 0.920	
**Histology** Ductal Lobular Other	1 0.70 [0.21–2.27] 0.88 [0.27–2.85] *p* = 0.810	
**Adjuvant chemotherapy** No Yes	1 0.34 [0.19–0.62] ***p* = 0.004**	1 0.29 [0.16–0.55] ***p* < 0.001**
**Basal-like phenotype** Yes No	1 1.20 [0.67–2.17] *p* = 0.541	
**AR** AR+ AR−	1 0.62 [0.31–1.24] *p* = 0.163	
**AR/Cath-D co-expression** AR+/Cath-D+ Other profiles	1 0.58 [0.31–1.09] ***p* = 0.086**	1 0.49 [0.25–0.96] ***p* = 0.034**
**TIL density** [0–2] 3	1 0.54 [0.24–1.21] *p* = 0.111	
**PDL-1 expression in tumor cells** <1% ≥1%	1 0.91 [0.49–1.70] *p* = 0.770	
**PD-1 expression in TILs** 0 1 2 3	1 0.69 [0.25–1.95] 1.05 [0.43–2.57] 0.72 [0.22–2.38] *p* = 0.689	
**Cath-D expression in tumor cells** No (score 0/+) Yes (score ++/+++)	1 1.28 [0.57–2.88] *p* = 0.534	
**Macrophages** 0/+1 2 3	1 1.12 [0.51–2.44] 0.64 [0.30–1.38] *p* = 0.219	

HR = hazard ratio; CI = confidence interval; *p* Value in bold, statistically significant.

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
