# Peer review of "Co-Expression of Androgen Receptor and Cathepsin D Defines a Triple-Negative Breast Cancer Subgroup with Poorer Overall Survival"

_cancers, 2020, doi:10.3390/cancers12051244_

Round 1

Reviewer 1 Report

In this manuscript the authors found the co-expression of androgen receptor and cathepsin D in triple-negative breast cancer subgroup and the co-expression is associated with poor overall survival. It's an interesting work. However, I am more interested in the androgen antagonists and anti-Cath-D combination treatment results.

The crucial experiment need to be done: How androgen antagonists and anti-Cath-D combination treatment affect the TNBC cell lines? (proliferation, migration, apoptosis?)

Reviewer 2 Report

Triple-negative breast cancers (TNBC) are molecularly heterogenous and lack effective targeted therapy. AR-positive TNBC has similar molecular profiles to ER-positive breast cancers. Androgens promote growth and proliferation of this type of TNBC. Therefore, clinical trials using antiandrogen therapy is currently under way to treat this type of TNBC. This study by Mansouri et al examined coexpression of AR /Cathepsin D in TNBC using breast cancer tissue microarrays and also evaluated whether this coexpression could be correlated with patient overall survival by multivariate analyses. Of the 147 non-metastatic TNBC patients analysed, 62.7% expressed both AR and Cath-D, and this co-expression was associated with worse overall survival. Incubation with enzalutamide had no effect on Cath-D expression. This study provided preliminary data supporting a combinatorial therapy for AR/Cath D positive TNBC by targeting AR and Cath D simultaneously. Although this study was novel, evidence from molecular cell biology supporting the combinatorial therapy is lacking as below.

 #1 This study showed no impact of antiandrogens on Cath-D expression. Did they examine the impact of the human anti-Cath-D F1 antibody on AR expression?

#2 AR/Cath-D coexpression was correlated with worse patient survival, this does not necessarily mean that this coexpression can promote TNBC growth. Evidence from cell lines or xenograft models are necessary to demonstrate that this coexpression stimulate TNBC growth and that targeting both genes simultaneously can inhibit TNBC growth and proliferation.

Reviewer 3 Report

Overall of the manuscript is interesting.

  1. One major concern is the unequal stage of samples in each group. In androgen positive group had higher nodal status. There was also unbalance number for each arm. In multivariate analysis, the nodal status should be in the model, because this factor was showed significant. Author should state all limitations in the discussion.
  2. Methodology part was absent.

Minor issues: Androgen receptor and prognosis of breast cancer is controversial, there are both sizes of pros and cons. Author should state about this issue in the introduction part.  

Round 2

Reviewer 1 Report

Accept in present form

Author Response

no response

Reviewer 2 Report

The authors did not address my comments experimentally for evidence supporting the "eligible for targeted combinatorial therapy". Therefore, this study only reported "Co-expression of androgen receptor and cathepsin D defines a triple-negative breast cancer subgroup with poorer overall survival", but lacked adequate evidence supporting " the eligible for targeted combinatorial therapy". To reflect this fact, it is required to remove "eligible for targeted 4 combinatorial therapy" from the title. It was reasonable to discuss this possibility in the discussion section. 

Author Response

As asked by the Reviewer 2, we removed "eligible for targeted combinatorial therapy" from the title.

Reviewer 3 Report

Lymph node status is one of the prognostic factor of breast cancer; otherwise we do not need TNM staging.

Author Response

Point 1. Lymph node status is one of the prognostic factor of breast cancer; otherwise we do not need TNM staging.

Response 1. In univariate analyses, nodal status is statistically correlated with recurrence-free survival and overall survival. In multivariate analyses, nodal status is not correlated with recurrence-free and overall survival, also nearly statistically significant (p=0.074 and 0.082, respectively).